# Designing infographics in health research with patients and the public: A scoping review protocol

**Blaze Beecher**[1], **Alan O'Doherty**[1], **Beatriz Goulao**[2], **Amirhossein Jalali**[1,3], **Jon Salsberg**[1,3], **Liz Dore**[4], **Ailish Hannigan**[1,3]*

**1** School of Medicine, University of Limerick, Limerick, Ireland, **2** Health Services Research Unit, University of Aberdeen, Aberdeen, United Kingdom, **3** Health Research Institute, University of Limerick, Limerick, Ireland, **4** Glucksman Library, University of Limerick, Limerick, Ireland

* ailish.hannigan@ul.ie

**Data Availability Statement:** No datasets were generated or analysed during the current study. All

## Abstract

Information graphics or infographics combine visual representations of information or data with text. They have been used in health research to disseminate research findings, translate knowledge and address challenges in health communication to lay audiences. There is emerging evidence of the design of infographics with the involvement of patients and the public in health research. Approaches to involvement include public and patient involvement, patient engagement and participatory research approaches. To date, there has been no comprehensive review of the literature on the design of infographics with patients and the public in health research. This paper presents a protocol and methodological framework for a scoping review to identify and map the available evidence for the involvement of patients and the public in infographics design in health research. It has been informed by preliminary searches and discussions and will guide the conduct and reporting of this review.

## Introduction

Information graphics or infographics combine visual representations of information or data with text "to communicate complex evidence-based information in an attractive and easily understandable way" [1]. Infographics can be static, animated or interactive and seek to educate, inform, or persuade the target audience [2]. In health research, they have been used to disseminate research findings, to translate knowledge and to increase health-promoting behaviours. Infographics may also address challenges in health communication to lay audiences with lower levels of health literacy or language barriers [3].

The G.R.A.P.H.I.C guidelines provide seven principles for the design of public health infographics, the first of which is to get to know your audience [4]. Scott et al. recommends getting advice from the target population group to ensure the infographic is appealing and understandable to them [5]. Piloting and evaluating the infographic on the target audience is recommended to verify that "the soul of the infographic" reaches them [6]. Given the central importance of the target population, design strategies for infographics which actively involve

relevant data from this study will be made available upon study completion.

**Funding:** The authors received no specific funding for this research.

**Competing interests:** The authors have declared that no competing interests exist.

**Table 1. Terminology and definitions.**

| Terminology | Definition |
| --- | --- |
| Public and Patient Involvement | "Research being carried out 'with' or 'by' members of the public rather than 'to', 'about' or 'for' them." [10] |
| Patient Engagement | "The active, meaningful, and collaborative interaction between patients and researchers across all stages of the research process, where research decision making is guided by patients' contributions as partners, recognizing their specific experiences, values, and expertise." [11] |
| Co-production | "An approach in which researchers, practitioners and the public work together, sharing power and responsibility from the start to the end of the project, including the generation of knowledge." [12] |
| Participatory Health Research | "The goal of Participatory Health Research is to maximize the participation of those whose life or work is the subject of the research in all stages of the research process." [13] |

patients and the public may be useful. Approaches to involvement include public and patient involvement, patient engagement and participatory research approaches (see Table 1 for definitions).

There is emerging evidence of the design of infographics with the involvement of patients and the public in health research. Arcia et al. used iterative participatory co-design sessions with Hispanic family caregivers of people with dementia to identify which infographic proto-types supported caregivers' comprehension of health status [7]. Staatz et al. used a public and patient involvement approach with pregnant women to develop an infographic to promote a healthy diet during pregnancy, providing the women with the choice of two aesthetically different options and getting input on their design, content, missing information and ease of understanding [8]. Campbell et al. used patient engagement techniques with children who had experienced a concussion and their parents to develop, refine and evaluate the usability of an education infographic on paediatric concussion [9].

To date, there has been no comprehensive review of the literature on the design of infographics with patients and the public in health research. The objective of this scoping review is to address this gap by identifying and mapping the available literature. Scoping reviews are useful for mapping an emerging body of literature by methods used, key concepts and characteristics, and types of evidence [14]. They often identify a broader range of evidence, acting as a precursor for systematic reviews and can also identify knowledge gaps [14, 15]. This protocol outlines the steps we will use to carry out this review, informed by best practice guidance and reporting for the development of scoping review protocols [15].

## Materials and methods

This protocol has been reported using the Preferred Reporting Items for Systematic Reviews and Meta-analyses extension for systematic review protocols (PRISMA-P) [16] S1 Checklist.

### Methodological framework

This scoping review will use Arksey and O'Malley's methodological framework [17], adapted by Levac et al. [18], with six stages: (i) Identifying the research question; (ii) Identifying relevant studies; (iii) Study selection; (iv) Charting the data; (v) Collating, summarising, and reporting results; and (vi) Consultation. The results of the review will be reported using the Preferred Reporting Items for Systematic Reviews and Meta-analyses extension for scoping reviews (PRISMA-ScR) [19].

## Stage 1: Identifying the research question

We used the Population Concept Context framework to develop our review question. The population is patients and the public, the concept is designing infographics and the context is health research. Our review question is 'How have patients and the public been involved in designing infographics in health research?'

## Stage 2: Identifying relevant studies

**Eligibility criteria.** *Population*. All studies with patients and the public are eligible for inclusion without restriction on demographics or diagnosis. Studies with non-human participants will be excluded. Studies with patients and the public, together with other participants e.g. researchers, healthcare practitioners or healthcare students, will be included only if the data for patients and the public can be disaggregated from the results.

*Concept*. All studies which use approaches for involvement (participatory, public and patient involvement, patient engagement) to develop an infographic will be included. This will include, for example, co-design with patients and the public; usability and comprehension testing; evaluation of acceptability and satisfaction; and feedback on content and presentation format by patients and the public. Our focus is specifically on infographics, excluding video assisted education and visual narratives. Studies without any approaches for involvement and without the development of an infographic will be excluded.

*Context*. All health research studies are eligible for inclusion without restriction on health topic, study design, setting (e.g. community, healthcare) or geographical location. A broad definition of health will be used to encompass physical, mental and social well-being [20]. Studies without a focus on health will be excluded.

*Types of sources*. All study designs (quantitative, qualitative, and mixed methods) will be included. Reviews of the literature will be included. Primary sources will be excluded if the data they contain have already been incorporated in an included review of the literature. Peer-reviewed and grey literature (conference proceedings, theses, reports) published in English (the language of the reviewers) with no date restriction are included. Protocols will be excluded.

Comprehensive searches will be carried out in the following databases: Scopus, PubMed Central, Web of Science Core Collection, EMBASE, CINAHL Complete, PsychINFO, Cochrane Library and ProQuest Dissertations and Theses. Google Scholar will be used for searching for relevant grey literature. Reference lists of all included studies will be screened for any additional relevant studies. Key authors will be contacted to provide potentially relevant studies for review.

Examples of preliminary database searches conducted in Web of Science and Scopus are given in Table 2 using title, abstract, and keywords fields. The search strategy will be adapted for each database and further refined in consultation with a research librarian. Any changes to the strategy will be documented and reported. All searches will be conducted after approval of the protocol.

## Stage 3: Study selection

Citations identified in the search will be compiled and exported to Endnote 20 (Clarivate Analytics, PA, USA) bibliographic software. Duplicates will be removed. Citations will be imported into Covidence systematic review software (Veritas Health Innovation) for title and abstract screening. The screening process will be independently piloted by two reviewers on a subset of 50 citations and then all citations will be screened, by title and abstract, using the inclusion and exclusion criteria. Full-texts of potentially relevant citations will be sourced and reviewed by two independent reviewers with disagreement resolved by discussion or by a third reviewer. Reasons for exclusion at full-text review will be documented. The results of the identification,

**Table 2. Examples of preliminary database searches.**

| |
|---|
| Database: Web of Science Core Collection (searching 'topic' field: title, abstract, author keywords, and Keywords Plus). |
| Date: 13/04/23 |
| Infographic* AND (involve* OR participat* OR engag* or co-creat* OR co-design*) AND (patient OR public OR consumer) (Topic) and English (Languages) |
| Results: 143 documents |
| Database: Scopus |
| Date 13/04/23 |
| ( TITLE-ABS-KEY ( infographic* ) AND TITLE-ABS-KEY (involve* OR participat* OR engag* or co-creat* OR co-design*) AND TITLE-ABS-KEY (patient OR public OR consumer ) ) AND ( LIMIT-TO ( LANGUAGE , "English" ) ) |
| Results: 193 documents |

screening, eligibility and inclusion of citations will be presented in a PRISMA-ScR flow diagram [19].

## Stage 4: Charting the data

The data extracted from each publication will include general information on the publication (e.g. authors, year of publication), details about the study (e.g. methods, location), details on the population, concept, context, and key findings relevant to the review. A draft data extraction tool is given in Table 3. This will be piloted by two independent reviewers on a subset of

**Table 3. Draft data extraction tool.**

| Categories | Questions |
|---|---|
| **Publication Details** | |
| Author(s) | Who are the authors of the publication? |
| Year of Publication | When was it published? |
| Country of origin | Where was the study carried out? |
| Publication Type | Is the publication a journal article, conference proceeding, thesis, report, other? |
| **General Overview of Study** | |
| Objective and aims | What was the objective and aims of the study? |
| Methods | What methods (e.g. qualitative, quantitative, mixed methods) were used to design/evaluate the infographic? |
| | What methods were used for involvement of patients and the public in the design process? |
| **Population** | Who were the participants e.g. how were they recruited, number of participants and their demographics? |
| **Concept** | What was the purpose of the infographic e.g. to education, inform, persuade? |
| | How was involvement defined? |
| | What was the role of the participants e.g. choosing between prototypes, deciding on content, assessing understanding, interpreting data, as co-authors? |
| **Context** | What was the health topic? |
| | Who did the infographic target and in what setting e.g. community, healthcare? |
| **Findings/results** | Was the infographic evaluated and if so, how? |
| | What were the outcomes? |
| | Was the role of patients and the public evaluated and if so, how? |
| | What changes were made to the infographic as a result of the involvement of patients and the public? |
| | Any challenges/limitations reported? |

10 included publications. The draft tool will be adapted as needed during the data extraction process, in discussion with the two independent reviewers and a third reviewer if needed. Any changes to the tool will be documented and reported. Authors of publications will be contacted if all data to be extracted is not available in the published paper.

### Stage 5: Collating, summarising and reporting the results

Scoping reviews, unlike systematic reviews, do not typically assess methodological quality or risk of bias of included studies or conduct data synthesis e.g. a meta-analysis [15]. Instead, a descriptive summary of included studies is provided. We will numerically and narratively summarise year of publication, geographical location, setting, study design, health topic, and purpose of the infographic in included studies. This will be used to report the most common areas of application of involvement, whether there is evidence of increasing use of involvement of patients and the public over time and whether it is more common for particular study designs and countries. A descriptive numerical summary of the number of participants and their demographics will also be conducted to identify the range and typical number of participants and common characteristics. In addition, a narrative summary of the process and evaluation of designing infographics will be used to answer the review question on how patients and the public have been involved. Outcomes of the evaluation of the infographic will be reported and summarised. Findings will be organised into thematic categories relating to types of studies, process and evaluation of involvement, and research gaps.

### Stage 6: Consultation

As recommended by Levac et al., we will include consultation as a necessary component of the proposed scoping review [18]. We will use the consultation stage to gain new perspectives on and meaning of preliminary findings, discuss potential for application and dissemination of findings and identify areas for future research. We plan consultation with the Public and Patient Involvement Research Unit, a multidisciplinary unit in the University of Limerick with collaborative partnerships with community organisations, academia, health service providers and policy makers to build evidence about meaningful public and patient involvement.

### Discussion

This protocol outlines the methodological framework and process we will use to identify and map the available evidence on the involvement of patients and the public in the design of infographics in health research. It has been informed by discussion and preliminary searches and will be used to guide the conduct and reporting of this review. Publishing a review protocol can reduce research waste, increase transparency and avoid selective reporting of results [15]. It also provides an opportunity for feedback from the research community in advance of the review which can provide valuable input to inform its conduct.

### Supporting information

**S1 Checklist. PRISMA-P checklist.**
(DOCX)

### Author Contributions

**Conceptualization:** Beatriz Goulao, Ailish Hannigan.

**Methodology:** Blaze Beecher, Alan O'Doherty, Beatriz Goulao, Amirhossein Jalali, Liz Dore, Ailish Hannigan.

**Supervision:** Ailish Hannigan.

**Writing – original draft:** Ailish Hannigan.

**Writing – review & editing:** Blaze Beecher, Alan O'Doherty, Beatriz Goulao, Amirhossein Jalali, Jon Salsberg, Liz Dore, Ailish Hannigan.

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
