## [Decision Letter · Decision Letter 0]

22 Aug 2023

Designing infographics in health research with patients and the public: a scoping review protocol

PONE-D-23-18262

Dear Dr. Ailish Hannigan,

We’re pleased to inform you that your manuscript has been judged scientifically suitable for publication and is formally accepted for publication .

Kind regards,

AKM Alamgir, PhD

Academic Editor

PLOS ONE

Journal Requirements:

1. Thank you for stating the following financial disclosure: 

"The authors received no specific funding for this research."

e) Please provide an amended Funding Statement that declares *all* the funding or sources of support received during this specific study (whether external or internal to your organization) as detailed online in our guide for authors at http://journals.plos.org/plosone/s/submit-now.  

f) Please state what role the funders took in the study.  If any authors received a salary from any of your funders, please state which authors and which funder. If the funders had no role, please state: "The funders had no role in study design, data collection and analysis, decision to publish, or preparation of the manuscript." 

Please send your amended statements by return email; we will change the online submission form on your behalf. 

Academic Editor's comments: The reviewers recommended this article to publish in our journal.

Reviewers' comments:

Reviewer's Responses to Questions

**Comments to the Author**

1. Does the manuscript provide a valid rationale for the proposed study, with clearly identified and justified research questions?

Reviewer #1: Yes

Reviewer #2: Yes

2. Is the protocol technically sound and planned in a manner that will lead to a meaningful outcome and allow testing the stated hypotheses?

Reviewer #1: Yes

Reviewer #2: Yes

3. Is the methodology feasible and described in sufficient detail to allow the work to be replicable?

Reviewer #1: Yes

Reviewer #2: Yes

4. Have the authors described where all data underlying the findings will be made available when the study is complete?

Reviewer #1: Yes

Reviewer #2: No

5. Is the manuscript presented in an intelligible fashion and written in standard English?

Reviewer #1: Yes

Reviewer #2: Yes

6. Review Comments to the Author

You may also provide optional suggestions and comments to authors that they might find helpful in planning their study.

Reviewer #1: Thank you for the opportunity to review your protocol As someone interested and invested in creating good quality co-created infographics I found this really interesting.

This seems to me to be a well constructed scoping review protocol focusing on a topic of interest and worth publishing.

A couple of points (and apologies if I have missed your mention of these). These are points that could be addressed.

You note in the introduction that the GRAPHIC guidelines provide seven principles.... but then don't mention them again. Are these embedded in someway in your data extraction sheet? It might be worth mentioning this.

Also, it may be that you intend to do this and I'be not properly noticed this in the data extraction sheet/protocol but will you be looking at the actual infographics yourself?

Also will you be reporting on the type of infographic (paper, electronic, static/animated etc.)?

Reviewer #2: The scoping review protocol is described in a clear and easy to follow format. I would be interested in seeing if you have done a pilot using this review methodology and if so were there any changes / adaptations made after this?

7. PLOS authors have the option to publish the peer review history of their article (what does this mean?). If published, this will include your full peer review and any attached files.

Reviewer #1: **Yes: **Bernie Carter

Reviewer #2: No

---

## [Editor Report · Acceptance letter]

25 Aug 2023

PONE-D-23-18262 

Designing infographics in health research with patients and the public: a scoping review protocol 

Dear Dr. Hannigan:

I'm pleased to inform you that your manuscript has been deemed suitable for publication in PLOS ONE. Congratulations! Your manuscript is now with our production department. 

Kind regards, 

on behalf of

Dr AKM Alamgir 

Academic Editor

PLOS ONE